# Use of a Small Car-Mounted Magnetic Resonance Imaging System for On-Field Screening for Osteochondritis Dissecans of the Humeral Capitellum

**DOI:** 10.3390/diagnostics12102551

**Published:** 2022-10-20

**Authors:** Kazuhiro Ikeda, Yoshikazu Okamoto, Takeshi Ogawa, Yasuhiko Terada, Michiru Kajiwara, Tomoki Miyasaka, Ryuhei Michinobu, Yuki Hara, Yuichi Yoshii, Takahito Nakajima, Masashi Yamazaki

**Affiliations:** 1Department of Orthopedic Surgery, Kikkoman General Hospital, Noda 278-0005, Chiba, Japan; 2Department of Orthopedic Surgery, Faculty of Medicine, University of Tsukuba, Tsukuba 305-8577, Ibaraki, Japan; 3Institute of Clinical Medicine, Department of Diagnostic and Interventional Radiology, University of Tsukuba, Tsukuba 305-8577, Ibaraki, Japan; 4Department of Orthopedic Surgery, Mito Medical Center, Ibaraki 311-3193, Ibaraki, Japan; 5Institute of Applied Physics, University of Tsukuba, Tsukuba 305-8577, Ibaraki, Japan; 6Department of Orthopedic Surgery, National Center of Neurology and Psychiatry, Kodaira 187-8551, Tokyo, Japan; 7Department of Orthopedic Surgery, Tokyo Medical University Ibaraki Medical Center, Ami 300-0395, Ibaraki, Japan

**Keywords:** magnetic resonance imaging, mobile, portable, osteochondritis dissecans, OCD, baseball, medical check-up, screening, low-field MRI

## Abstract

Mobile magnetic resonance imaging (MRI) using a car is a recent advancement in imaging technology. Specifically, a car-mounted mobile MRI system is expected to be used for medical check-ups; however, this is still in the research stage. This study demonstrated the practicality of a small car-mounted mobile MRI in on-field screening for osteochondritis dissecans (OCD) of the humeral capitellum. In the primary check-up, we screened the throwing elbows of 151 young baseball players using mobile MRI and ultrasonography. We definitively diagnosed OCD at the secondary check-up using X-ray photography and computed tomography or MRI. We investigated the sensitivity and specificity of mobile MRI and ultrasonography for OCD. Six patients were diagnosed with OCD. The sensitivity was 83.3% for mobile MRI and 66.7% for ultrasonography, with specificity of 99.3% vs. 100%, respectively. One patient was detected using ultrasonography but was missed by mobile MRI due to poor imaging quality at the first medical check-up. Following this false-negative case, we replaced a damaged radio frequency coil to improve the image quality, and the mobile MRI could detect all subsequent OCD cases. Two patients were diagnosed by mobile MRI only; ultrasonography missed cases lacking subchondral bone irregularity, such as a healing case, and an early-stage case. Mobile MRI could screen for OCD from the very early stages through the healing process and is therefore a practical tool for on-field screening.

## 1. Introduction

The use of mobile imaging tools is increasingly considered an effective way of utilizing limited medical resources [1,2,3]. Moreover, mobile imaging enables qualified medical personnel to practice remotely. It is currently widely applied for medical check-ups at workplaces or schools, and the time and cost reduction for examinees is expected to increase the examination rate [4].

In the bone and soft tissue field, mobile imaging tools play an important role in athletic medical check-ups [5,6,7,8,9]. Osteochondritis dissecans (OCD) of the humeral capitellum is a bone and cartilage disorder for which athletic medical check-ups are useful. OCD occurs in young baseball players aged 9–12 years, with a prevalence of 2.1–3.4% [7,10]. Early detection is essential to prevent elbow dysfunction; 90% of patients with stage I OCD recover with conservative treatment, whereas 50% of patients with stage II OCD require surgery [11]. Since the early stage of OCD is often asymptomatic [12,13], a medical check-up is essential for early detection. Moreover, on-field screening is generally performed in Japan because it enables the examination of all players of the team. Currently, the gold standard imaging examination for on-field screening is ultrasonography (US) [7,10,12,13] due to its advantages, including portability, short examination time, no radiation exposure, and low cost. In addition, US detects the OCD lesion of subchondral bone irregularity from its early stage [7,10,12,13], making it an excellent qualitative examination tool for OCD detection. 

Conversely, magnetic resonance imaging (MRI) is the gold standard for in-hospital diagnosis and evaluation of OCD lesions. MRI visualizes OCD lesions of the subchondral bone as low signal changes on T1-weighted images (T1WI) from the early stage, which cannot be detected by X-ray photography (X-p), computed tomography (CT) and US [14,15,16]. MRI detects qualitative changes in bone marrow edema, whereas X-p, CT, and US detect morphological changes and subchondral bone irregularities, as in other occult fractures [17]. MRI is therefore better suited to detect early cases than other imaging modalities. Although MRI has never been used for on-field OCD screening because it was not portable, introducing mobile MRI into OCD screening may help detect earlier cases.

However, the mobilization of MRI is still at the research stage, in contrast to X-p and US. As a result of prioritizing high resolution, current mobile MRI systems are equipped with 1.5 T magnets and require a 38-ton trailer [18,19]. Such a large mobile MRI truck is expensive, must be parked on a level area with reinforced pads, and requires a 480 V 3-phase power supply [3]; this mobile MRI truck has a high threshold for introduction and operation. Therefore, we proposed that a reduction in size and cost was essential for widespread use of mobile MRI, and a low-field MRI could realize this [20,21]. Using low-field MRI, we developed a small car-mounted mobile MRI system for human small joints [1]. This system, comprising a 0.2 T permanent magnet, enabled us to perform MRI anywhere using a 100 V external power supply. We have previously demonstrated excellent imaging using this mobile MRI system with healthy volunteers [1].

In this study, we introduced mobile MRI into OCD screening. This study aimed to examine the practicality and utility of this mobile MRI in on-field screening for OCD of the humerus (Figure 1).

## 2. Materials and Methods

### 2.1. Study Design and Participants

This prospective, non-randomized observational study protocol conforms to the principles of the 1964 Declaration of Helsinki and its later amendments. The review board of our institution approved this study (IRB No: 30–144, approved 23 July 2019). Written informed consent was obtained from all participants. We evaluated 151 throwing-elbows of 151 young baseball players (149 boys and two girls; mean age 11.6 years; age range 8–15 years) who participated in our baseball medical check-up from October 2019 to May 2021. The median number of players per screening was 16 (8–19) among 10 teams.

### 2.2. The Mobile MRI system

The mobile MRI system consisted of a permanent magnet, gradient coils, a radio frequency (RF) probe with shielding cloths, and the MRI console (Figure 2). Since the RF probes are consumable, we replaced the damaged coil once during the study period; 32 cases were imaged with the first-generation RF probe, and 119 cases with the second-generation RF probe. The overall weight of the system was below the maximum authorized payload, and all the devices could be mounted in the vehicle. All the electronic devices were operated at 100 V AC, and the required current was 10 A. A power cable was connected from the vehicle to a wall outlet at the nearest building. Hence, we could perform MRI anywhere. Our mobile MRI system is comprehensively described in our previous paper [1].

### 2.3. OCD Screening

We conducted the primary medical check-up at the baseball field during the practice session of the target teams. In addition, we performed physical examinations, including the elbow range of motion (ROM), tenderness to palpation, and moving valgus test. Finally, we screened for OCD using the mobile MRI and US independently.

Once in the mobile MRI car, the participants sat on a legless chair and placed their forearm in the scanner, with the elbow joint in extension and the forearm supinated. The routine sequence images obtained were the sagittal elbow with T1WI and T2*-weighted imaging (T2*WI) and the coronal elbow with T2*WI (Figure 3 and Table 1). A musculoskeletal radiologist with 25 years’ experience (Examiner A) read the images immediately and determined the presence of OCD.

We can clearly identify bone morphology with T1WI sagittal images and articular cartilage morphology with T2*WI sagittal and coronal images.

We used a Viamo SV 7 scanner with a 3–10 MHz linear probe (Canon Medical Systems, Tochigi, Japan) for US examinations. The player sat on the chair and placed the arm on the table with the forearm supinated. We obtained a posterior longitudinal and short-axis view with the elbow fully flexed and an anterior longitudinal and short-axis view with the elbow fully extended [22,23,24]. Three orthopedic surgeons examined the US and determined the presence of OCD depending on the case: a hand surgeon with 23 years of experience (Examiner B) and two general orthopedic surgeons with 9 (Examiner C) and 8 (Examiner D) years of experience, respectively. All three examiners had at least 3 years of OCD screening experience.

We referred OCD-positive patients diagnosed with either mobile MRI or US for a secondary check-up. Consequently, all the OCD-positive patients underwent X-p, with computed tomography (CT) or MRI as an additional imaging examination. Based on the obtained images, a hand surgeon with 23 years of experience (Examiner E), who did not participate in the primary check-up, made a definitive diagnosis of OCD. A flow diagram of our medical check-up is shown in Figure 4.

### 2.4. Evaluation Components

The evaluation components in this study were as follows. First, the sensitivity and specificity of the mobile MRI and US. In calculating sensitivity and specificity, we defined true positive cases as the patients with a definitive OCD diagnosis in the secondary check-up, and true negative cases as the other patients. Second, the details of the OCD-positive cases, i.e., physical examination findings, position, OCD stage and location, and the maximum diameter of the lesion were evaluated. Third, we evaluated the OCD stage [11] and its location with an anteroposterior X-p of the elbow at 45° of flexion (Xp−45°). We also measured the maximum diameter of the lesion with CT or MRI. In CT, we defined the lesion as a subchondral bone irregularity or sclerosis. In MRI, we detected signal changes in the subchondral bone.

## 3. Results

Of the 151 players, 6 (4.0%) were diagnosed with OCD through our medical check-ups (Table 2). Three patients with OCD had no elbow pain and positive physical examination findings for OCD. Out of the six OCD cases, five tested positive on mobile MRI and four on US; two were positive only on mobile MRI, and one was positive only on US. One patient was OCD-positive on mobile MRI but was diagnosed with a different disorder at the secondary check-up. Consequently, the sensitivity was 83.3% for mobile MRI and 66.7% in US, with specificities of 99.3% and 100%, with positive predictive value of 83.3% and 100%, respectively (Table 2, Table 3 and Table 4). Legends of the cases are shown in Figure 5, Figure 6, Figure 7, Figure 8 and Figure 9.

The results of the secondary check-up are presented in Table 5. There were three cases of stage I OCD, three cases of stage II OCD, and one case with a different disorder. The two cases that were diagnosed only using mobile MRI included an early OCD case and a healing case. The case that was diagnosed only with US was stage II OCD, which occurred at the first medical check-up using mobile MRI. Representative cases are described in Figure 4, Figure 5, Figure 6, Figure 7 and Figure 8.

## 4. Discussion

This is the first report to show the clinical validity of a small car-mounted mobile MRI system. Our mobile MRI detected OCD cases from the very early stage through to the healing stage. Particularly in cases 5 and 7, mobile MRI detected OCD that was not detectable using US. One case was in the healing stage and the other was in the very early stage of OCD, without or with slight subchondral bone irregularity. Since US waves do not penetrate the subchondral bone, US cannot detect OCD without subchondral bone irregularity. In contrast, mobile MRI shows OCD-positive findings as distinct subchondral bone marrow edema. Mobile MRI has the potential to detect OCD cases without subchondral bone irregularities. Case 5 was in the healing process with a repaired subchondral bone surface. According to previous reports, shorter periods of conservative treatment tend to result in lower healing rates [11,24,25]. Thus, OCD in the healing stage is also a screening target. Case 7 was OCD at a very early stage. Although screening for early OCD before subchondral bone irregularities develop is ideal, this cannot be achieved using US. Therefore, from the perspective of early detection, mobile MRI is an excellent screening tool.

Meanwhile, there was a false negative case and false positive results with mobile MRI. The false negative case was due to poor imaging quality during the first medical check-up using the mobile MRI. The imaging quality of low-field MRI depends on various conditions, including the quality of the RF probe, imaging parameters, the patient’s motion, and the patient’s positioning in the coil [26,27,28,29]. In particular, RF probes are consumables, and the image quality deteriorates over time. Based on our experience of this false-negative case, we replaced the RF probe to increase its sensitivity and reduce noise, and improved the legless chair to reduce motion artifacts. Subsequently, the imaging quality was improved, and the mobile MRI could detect all OCD cases. Although case 7 was a false positive case, mobile MRI detected a small imaging abnormality with excellent imaging quality.

Regarding the practicality of mobile MRI for on-field OCD screening, we overcame some disadvantages of conventional mobile MRI. Our mobile MRI went to an unpaved baseball field and obtained MRI images of sufficient quality for diagnosis using a household power source. As a result, the MRI scan time was relatively short (1.5 min per sequence), and we could screen all team members within one practice session. Our future aim is to further reduce the examination time for screening a large number of players. We have continued to improve our RF probes to achieve this. If the imaging quality is high enough to diagnose OCD with only T1WI sagittal imaging, we can save a considerable amount of examination time. We will continue our research to detect as many early OCD cases as possible.

This study had some limitations. First, it only demonstrated the clinical use of mobile MRI and did not include any statistical analysis. In the future, we will recruit more cases to compare the diagnostic ability of mobile MRI and US. Additionally, multiple examiners performed the US examinations for participants. Although all examiners had adequate experience with US, the quality of these examinations might have been inconsistent because US proficiency could have varied [30]. Finally, we only performed secondary check-up for patients shown to be positive in the primary check-up, which may affect the sensitivity and specificity results. However, US has been established as the gold standard for OCD-screening in Japan. Therefore, we considered our study design appropriate to determine the accuracy of the diagnosis using mobile MRI.

## 5. Conclusions

Mobile MRI was used for on-field screening for OCD for an entire youth baseball team of around 20 players during one practice session. Furthermore, the imaging quality was sufficient to diagnose OCD; mobile MRI could screen for OCD from the very early stages through to the healing process, including cases without subchondral bone irregularity, which could not be detected using US. Therefore, mobile MRI is a practical tool for on-field screening and has the potential to screen all stages of OCD.

## Figures and Tables

**Figure 1 diagnostics-12-02551-f001:**
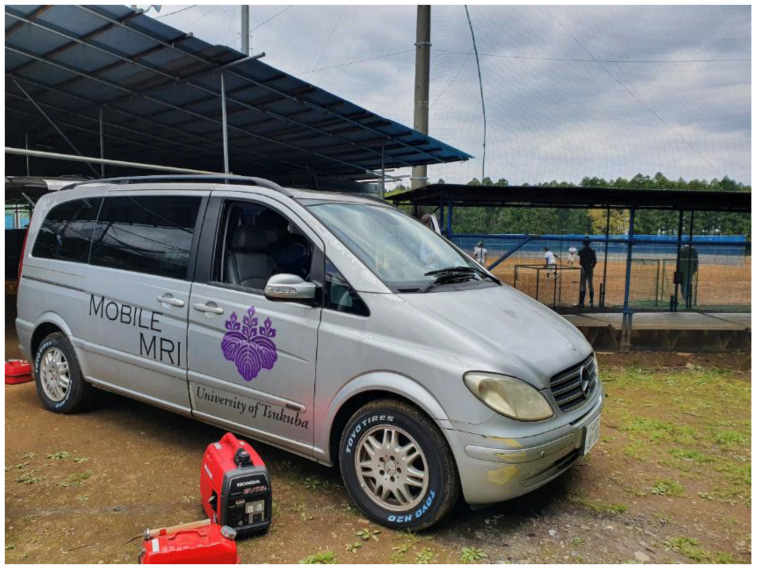
Photograph of the mobile magnetic resonance imaging (MRI) system beside a baseball field. (Mercedes Benz, GH-639811, width 191 cm, height 193 cm, and length 476 cm). Mobile MRI enables players to practice while awaiting their turn for an MRI scan.

**Figure 2 diagnostics-12-02551-f002:**
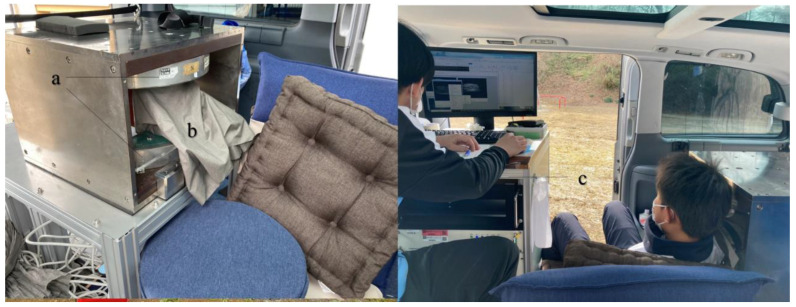
The mobile MRI system. (**a**) Permanent magnet. The magnet is a 0.2 T permanent magnet (NEOMAX Engineering, Japan; 200 kg; 16 cm-gap; 44 cm × 50 cm × 36 cm). The magnet was screwed onto an aluminum stand that was anchored to the sheet rail of the vehicle; (**b**) an RF probe was placed inside the conductive shielding cloths (ESD EMI Engineering Corporation, Tokyo, Japan). The home-built RF probe consisted of a solenoid RF coil (12 turns, 130 mm long, 94 mm in diameter) and a rectangular shield box (200 mm (x) × 200 mm (y) × 132 mm (z)) made of 200-µm-thick brass plates. The size of the RF coil was large enough to fit most of the junior baseball players; (**c**) MRI console: the MRI console consisted of a digital transceiver (DTRX6, MR Technology, Japan), a gradient driver (20 V, 10 A, DST Inc., Japan), a preamplifier (noise figure was 0.5 dB, gain was 30 dB; DST Inc., Asaka, Japan), an active transmit/receive switch, and a transmitter (9 MHz, 150 W; DST Inc.), which were installed on a 19-inch rack (56 cm × 77 cm × 60 cm, 80 kg). The MRI console was tightly fixed to the front seat with ropes.

**Figure 3 diagnostics-12-02551-f003:**
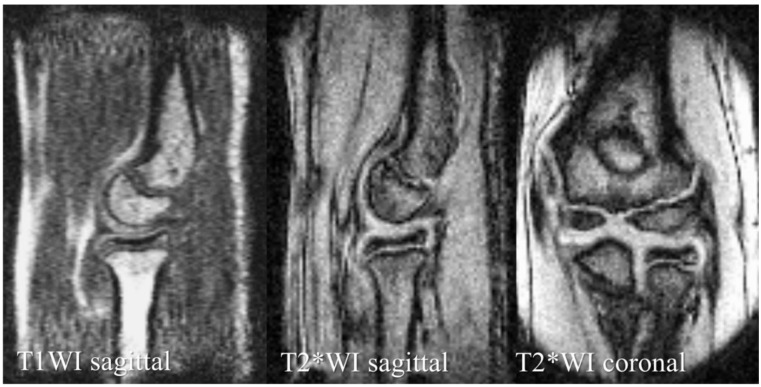
Mobile MRI of a healthy elbow of a 12-year-old boy.

**Figure 4 diagnostics-12-02551-f004:**
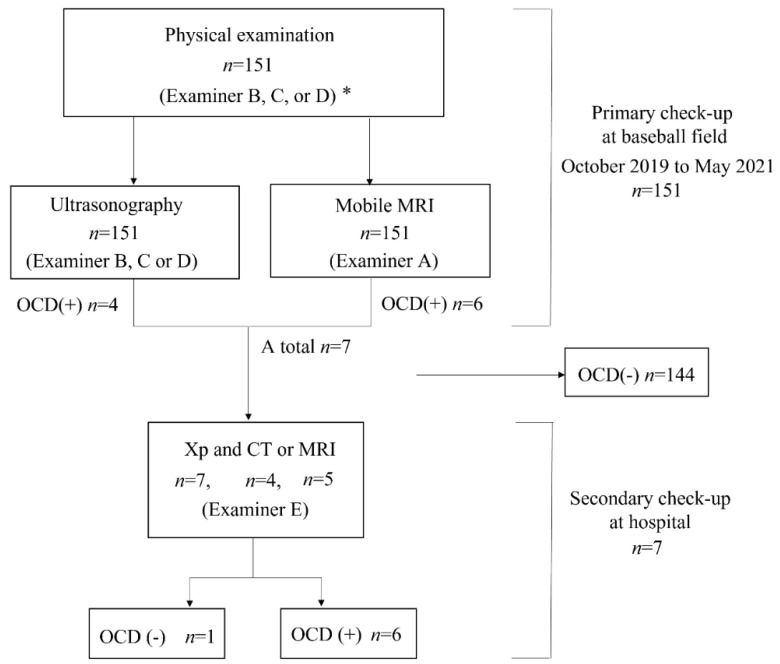
Flow diagram of our medical check-up. * Examiner A, a musculoskeletal radiologist with 25 years of experience; Examiner B, a hand surgeon with 23 years of experience; Examiner C, a general orthopedic surgeon with 9 years of experience; Examiner D, a general orthopedic surgeon with 8 years of experience; and Examiner E, a hand surgeon with 23 years of experience; OCD, osteochondritis dissecans of the humeral capitellum.

**Figure 5 diagnostics-12-02551-f005:**
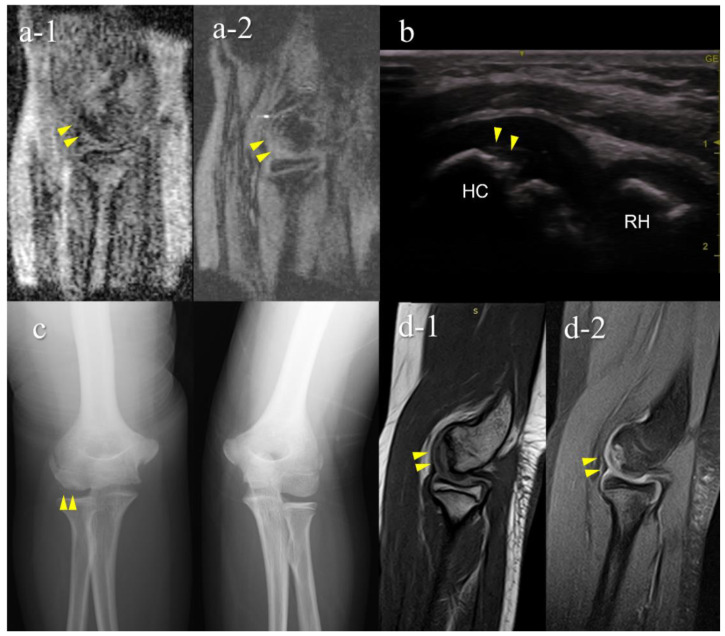
A false-negative OCD case with mobile MRI (Case 1): Mobile MRI (−), US (+). (**a**) Mobile MR (magnetic resonance) images: a-1, T1WI sagittal image; a-2, T2*WI sagittal image: Mobile MRI with poor imaging quality with low signal/noise ratio. Although the signal change was barely visible from the anterior to the humeral capitellum, we could not detect it; (**b**) US image of the posterior longitudinal view: US imaging revealed obvious subchondral bone irregularity; (**c**) X-p AP view with elbow 45° flexed: X-p 45° showing the lesion with fragmentation in the central capitellum, stage II OCD; (**d**) 3T MR images: d-1, proton density-weighted images (PDWI) sagittal image; d-2, PDWI- fat-suppressed (FS) sagittal image: 3T MRI easily detected the lesion. HC, humeral capitellum; RH, radial head; yellow arrowheads, lesions.

**Figure 6 diagnostics-12-02551-f006:**
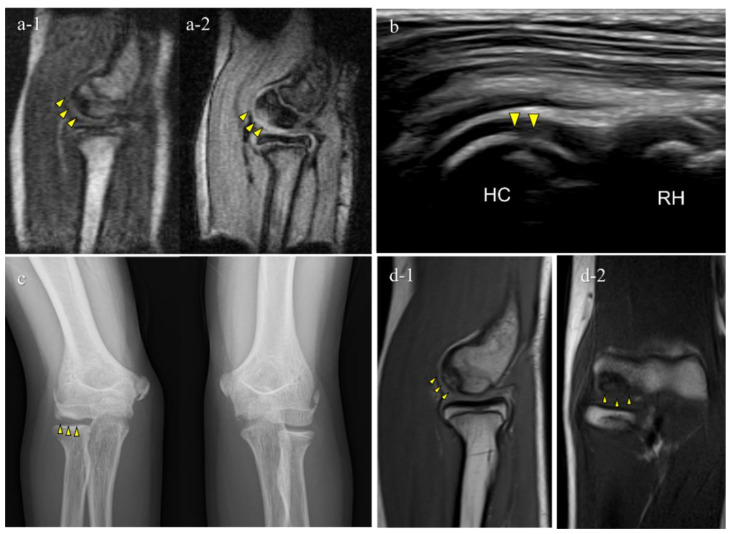
Typical OCD case (Case 3): Mobile MRI (+), US (+). (**a**) Mobile MR images; a-1, T1WI sagittal image; a-2, T2*WI sagittal image: Mobile MRI depicted the OCD lesion as a low-signal area in a T1WI sagittal image and a high-signal area in a T2* weighted image; (**b**) US image of the posterior longitudinal view: US showing subchondral bone irregularity; (**c**) X-p AP view with elbow 45° flexed: X-p 45° showing the lesion with fragmentation, stage II-OCD; (**d**) 3T MR images: d-1, PDWI sagittal image; d-2, coronal image: localization of the lesion was consistent with that of 3T MRI and mobile MRI.

**Figure 7 diagnostics-12-02551-f007:**
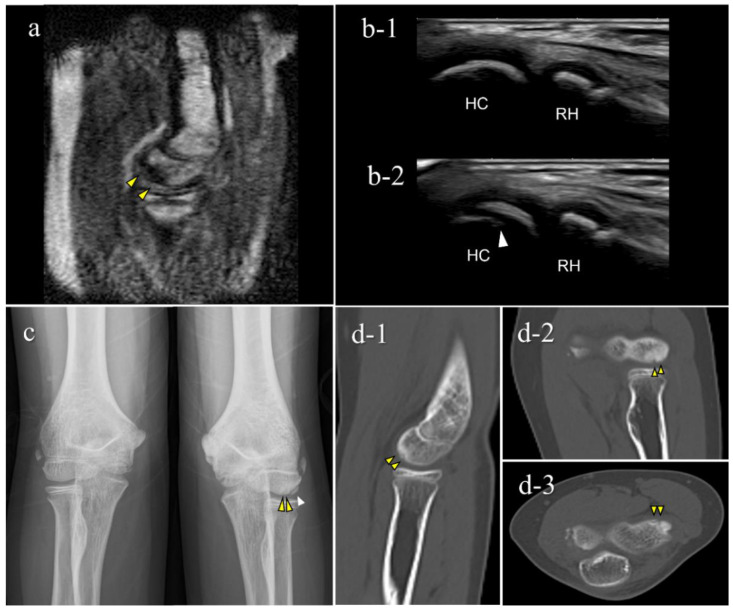
An OCD case during the healing process (Case 5): Mobile MRI (+), US (-). (**a**) Mobile MR image of T1WI sagittal: Mobile MRI showing signal intensity changes reflecting bone marrow edema in the anterior humeral capitellum; (**b**) US image of the posterior longitudinal view: b-1, the central posterior; b-2, the lateral posterior: US images showing no significant findings in the central posterior longitudinal view. However, there was a subchondral bone discontinuity in the lateral posterior longitudinal view; (**c**) X-p AP view with elbow 45° flexed: X-p 45° showing delayed ossification in the lateral capitellum, observed in incomplete healing OCD cases [11]. In addition, there was a translucent area in the central capitellum, stage I-OCD; (**d**) CT images: d-1, sagittal image; d-2, coronal image; d-3, axial image: CT images showing the subchondral bone sclerosis, and the surface was almost repaired. White arrowheads: delayed ossification.

**Figure 8 diagnostics-12-02551-f008:**
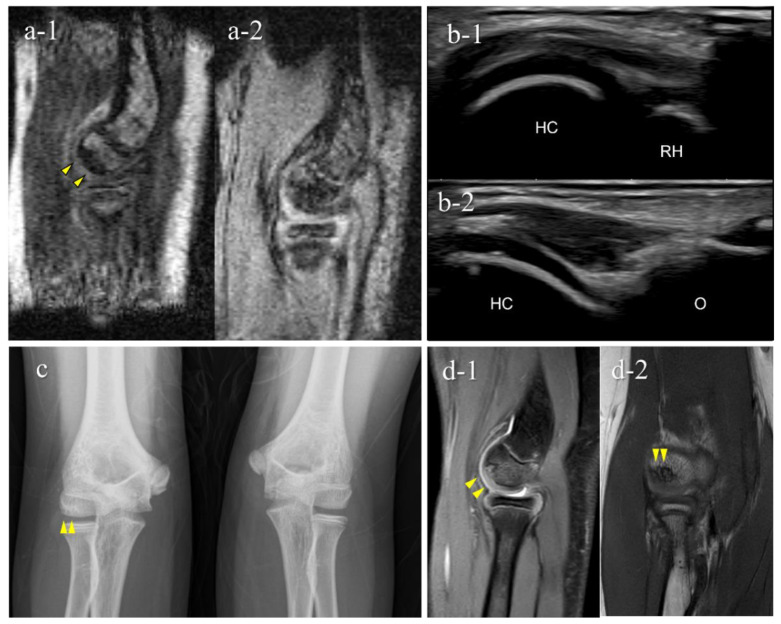
A very early stage of an OCD case (Case 7): Mobile MRI (+), US (−). (**a**) Mobile MR image of sagittal T1WI: Mobile MRI showing signal intensity changes reflecting bone marrow edema in the anterior humeral capitellum; (**b**) US image of the posterior longitudinal view: US did not show subchondral bone irregularity; (**c**) X-p AP view with elbow 45° flexed: X-p 45° showing a translucent area in the central capitellum, stage-I OCD; (**d**) 3T MR images: d-1, PDWI-FS sagittal image; d-2, PDWI coronal image: localization of the lesion was consistent with that of 3T MRI and mobile MRI.

**Figure 9 diagnostics-12-02551-f009:**
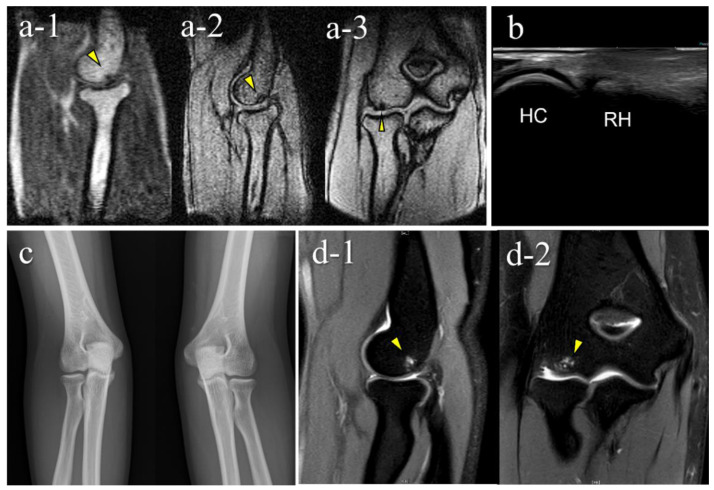
A false-positive case with mobile MRI (Case 6): Mobile MRI (+), US (-). (**a**) Mobile MR images: a-1, sagittal T1WI image; a-2, sagittal T2*WI image; a-3, coronal T2*WI image: mobile MR images showing a small subchondral bone-lesion in the posterior capitellum; (**b**) US images of the posterior longitudinal view: US did not reveal any remarkable findings; (**c**) X-p AP view with elbow 45° flexed: X-p 45° showed no significant findings; (**d**) 3T MR images: d-1, PDWI-FS sagittal image; d-2, PDWI-FS coronal image: 3-T MR image showing that the lesion was located in the posterior-central region of the capitellum and conflicted with the posterior synovial folds in the extended position. The pathogenesis differed from that of OCD caused by pitching according to the location of the lesion. Repeated hyperextension may have contributed to the pathology, as in Panner’s disease, based on the location of the lesion and the patient’s range of motion findings.

**Table 1 diagnostics-12-02551-t001:** Imaging parameters.

Sequence Type	T1WI	T2*WI	T2*WI
Image plane	Sagittal	Sagittal	Coronal
FOV	180 × 120 × 90 mm^3^	180 × 180 mm^2^	180 × 180 mm^2^
Matrix size	256 × 128 × 32	256 × 192	256 × 192
Slice thickness, mm	-	3	3
TR, ms	40	500	500
TE, ms	4	16	16
Flip angel	60°	75°	75°
Scan time	1 min 22 s	1 min 38 s	1 min 38 s

T1WI, T1 weighted images; T2*WI, T2* weighted images; FOV, field of view; TR, repetition time; TE, echo time; ms: millisecond.

**Table 2 diagnostics-12-02551-t002:** Results of the medical check-up.

	OCD (−)	OCD (+)
Cases	145	6
Mean age (years)	11.6 ± 1.3	12.3 ± 1.2
Sex	Male: 143	Male: 6
	Female: 2	Female: 0
Medial elbow pain, n (%)	30 (20.7)	3 (50.0)
Lateral elbow pain, n (%)	6 (4.2)	3 (50.0)
No elbow pain, n (%)	113 (77.9)	3 (50.0)

**Table 3 diagnostics-12-02551-t003:** Diagnostic accuracy of OCD using mobile MRI.

		Definitive Diagnosis		
		OCD (+)	OCD (−)	
mobile MRI	OCD (+)	5	1	6
	OCD (−)	1	144	145
		6	145	151

**Table 4 diagnostics-12-02551-t004:** Diagnostic accuracy of OCD using US.

		Definitive Diagnosis		
		OCD (+)	OCD (−)	
US	OCD (+)	4	0	4
	OCD (−)	2	145	147
		6	145	151

**Table 5 diagnostics-12-02551-t005:** Characteristics of OCD-positive players.

Case	Age	Sex	Primary Check-Up			Secondary Check-Up			
			Elbow Symptom	Mobile MRI (RF Probe- Generation)	US (Examiner)	Diagnosis	Stage	Lesion Location	Lesion Diameter (mm)
1	12	M	-	− (1st)	+ (B)	OCD	II	lateral	15
2	11	M	+	+ (1st)	+ (B)	OCD	I	central	10
3	12	M	-	+ (2nd)	+ (B)	OCD	II	central	9
4	12	M	+	+ (2nd)	+ (B)	OCD	II	lateral	11
5	12	M	+	+ (2nd)	− (B)	OCD	I	central	3.5
6	15	F	-	+ (2nd)	− (C)	not OCD		posterior	3.5
7	12	M	-	+ (2nd)	− (D)	OCD	I	central	5.5

US, ultrasonography, M, male; F, female; RF probe, radio frequency probe.

## Data Availability

Not applicable.

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
