# Peer review of "Use of a Small Car-Mounted Magnetic Resonance Imaging System for On-Field Screening for Osteochondritis Dissecans of the Humeral Capitellum"

_diagnostics, 2022, doi:10.3390/diagnostics12102551_

Round 1

Reviewer 1 Report

This manuscript provides a pilot study on the prospect of mobile MRI compared to US and MRI itself. Of course there will be advantage and disadvantage. Overall, this manuscript provides a good research outcome. I recommend an acceptance to this manuscript.

Author Response

This manuscript provides a pilot study on the prospect of mobile MRI compared to US and MRI itself. Of course there will be advantage and disadvantage. Overall, this manuscript provides a good research outcome. I recommend an acceptance to this manuscript.

Response) Thank you for your review and your recommendation of acceptance.

Reviewer 2 Report

In the abstract one should be probably say that "mobile magnetic resonance with a car" is a recent advance. Mobile MRI with a large truck (as they note later) is not recent and not distinguishing between car and truck might be seen as incorrect by some readers. 

Also in the abstract and body of the paper they should define X-p. 

On page 2 line 65 the authors should clarify what is meant by insufficient fractures. Line 69 they should change "huge mobile MRI car" to "truck". 

Page 3 figure 2 they refer to Figure 1(e) in the caption but it is not clear what they mean. There is no (e) in Figure 1. 

Page 7 they should say how they calculate specificity. 

Author Response

In the abstract one should be probably say that "mobile magnetic resonance with a car" is a recent advance. Mobile MRI with a large truck (as they note later) is not recent and not distinguishing between car and truck might be seen as incorrect by some readers. 

Response) Thank you for the comment. We have corrected the description in the abstract. (Page 1 Line 24)

Also in the abstract and body of the paper they should define X-p. 

Response) Thank you for pointing this out, we have added the definition for X-p. (Page 1 Line 30, and Page 2 Line 66)

On page 2 line 65 the authors should clarify what is meant by insufficient fractures. 

Response) Thank you for bringing this to our attention, we revised the description of insufficient fracture. (Page 2 Line 68)

Line 69 they should change "huge mobile MRI car" to "truck".

Response) Thank you for pointing this out. We have corrected the description. (Page 2 Line 76 and 77)

 Page 3 figure 2 they refer to Figure 1(e) in the caption but it is not clear what they mean. There is no (e) in Figure 1. 

Response) Thank you for careful review and highlighting this oversight. We removed description of "(e)". (Page 4 Line 124)

Page 7 they should say how they calculate specificity. 

Response) Thank you for pointing this out. We added the definition of true positive and negative cases in this study and the description for the specificity (Page6 Line 169-178). We performed secondary check-up only in patients shown to be positive in the primary check-up, which may affect the sensitivity and specificity results. We have added this as a limitation. Since US has been established as the gold standard for OCD-screening, we believe that our study design was appropriate to determine accuracy of the diagnosis using mobile MRI. (Page 14 Line 298-302)

Reviewer 3 Report

This paper introduces MRI to mobile diagnostics, improves on some of the shortcomings of introducing MRI to mobile devices, and explores the advantages of mobile magnetic resonance imaging (MRI) in screening for osteochondritis dissecans of the humeral capitellum by comparing it to the US approach. However, this paper needs more improvement. The specific modifications are as follows.

1. In the abstract, for the missed patients, the paper explains that the poor imaging quality of the initial test is the cause. So after improving the imaging, can the missed patients be detected accurately?

2. On the basis of comment 1, the number of patients should be increased for further testing.

3. In the introduction, the paper describes the need for early diagnosis of this disease. However, in addition to this, the current status of existing techniques for early diagnosis, their shortcomings, and the advantages of MRI should be listed.

4. In the results, more valid evaluation indicators should be included. For example, the way of handling false positives in addition to false negatives.

5. In the discussion, a discussion of cost should be included in addition to patient detection rate. This will better validate the feasibility of mobile MRI.

Author Response

This paper introduces MRI to mobile diagnostics, improves on some of the shortcomings of introducing MRI to mobile devices, and explores the advantages of mobile magnetic resonance imaging (MRI) in screening for osteochondritis dissecans of the humeral capitellum by comparing it to the US approach. However, this paper needs more improvement. The specific modifications are as follows.

  1. In the abstract, for the missed patients, the paper explains that the poor imaging quality of the initial test is the cause. So after improving the imaging, can the missed patients be detected accurately?

Response) Thank you for the comment. We added the following sentence in the abstract. “Following this false-negative case, we modified the radio frequency coil to improve image quality, and mobile MRI could detect all subsequent OCD cases.” (Page 1 Line 34-36)

  1. On the basis of comment 1, the number of patients should be increased for further testing.

Response) As you pointed out, the improvement in image quality due to the replacement of the RF probe affects the results of this study. Since it is difficult to compare detection rates with RF probes, we have included details of the descriptions for RF as follows. First, we noted that the RF coil is a consumable item and needs to be replaced as needed (Page3 Line102-103, Page14 Line276-277). Second, we added the number of cases examined with the first- and second-generation RF probes (Page3 Line104-105). Finally, we also added which probe was used to diagnose the positive cases (Page8, Table 1).

  1. In the introduction, the paper describes the need for early diagnosis of this disease. However, in addition to this, the current status of existing techniques for early diagnosis, their shortcomings, and the advantages of MRI should be listed.

Response) Thank you for your pointing this out. We added the following sentence in the introduction. “MRI detects qualitative changes of bone marrow edema, whereas X-ray, CT, and US detect morphological changes and subchondral bone irregularities, as in other occult fractures [17]. MRI is therefore better suited to detect early cases than other imaging modalities. Although MRI has never been used for on-field OCD screening because it was not portable, introducing mobile MRI into OCD screening may help detect earlier cases.” (Page 2 Line 67-72)

  1. In the results, more valid evaluation indicators should be included. For example, the way of handling false positives in addition to false negatives.

Response) Thank you for pointing this out, we added the positive predictive value in the results. (Page 7 Line 184-185) Additionally, we described the definition of true negative and positive cases in this study. (Page6 Line169-178)

  1. In the discussion, a discussion of cost should be included in addition to patient detection rate. This will better validate the feasibility of mobile MRI.

Response) Thank you for pointing this out. As you indicated, the cost is a key factor needs to be considered in this study. The cost of developing a mobile MRI is approximately 10 million yen (67,114 $), including the costs for the car, the console, and RF coil. However, developers and owners of this mobile MRI are reluctant to disclose development costs because of the fear of theft. Like regular automobiles, mobile MRIs are at risk of theft. Therefore, we would like to refrain from mentioning the cost in the text. We crave your understanding regarding this issue.

Round 2

Reviewer 3 Report

the work can be accepted